# Impact of COVID-19 Pandemic Exacerbation of Depressive Symptoms for Social Frailty from the ORANGE Registry

**DOI:** 10.3390/ijerph19020986

**Published:** 2022-01-16

**Authors:** Ayuto Kodama, Yu Kume, Sangyoon Lee, Hyuma Makizako, Hiroyuki Shimada, Tomoko Takahashi, Tsuyoshi Ono, Hidetaka Ota

**Affiliations:** 1Advanced Research Center for Geriatric and Gerontology, Akita University, Akita 010-8543, Japan; ay-kodama@med.akita-u.ac.jp; 2Department of Occupational Therapy, Graduate School of Medicine, Akita University, Akita 010-8543, Japan; kume.yuu@hs.akita-u.ac.jp; 3Center for Gerontology and Social Science, National Center for Geriatrics and Gerontology, Obu 474-8511, Japan; sylee@ncgg.go.jp (S.L.); shimada@ncgg.go.jp (H.S.); 4Department of Physical Therapy, School of Health Science, Faculty of Medicine, Kagoshima University, Kagosima 890-8580, Japan; makizako@health.nop.kagoshima-u.ac.jp; 5Integrated Community Support Center, Public Health and Welfare Department, City Hall of Yokote, Yokote 013-0525, Japan; Takahashi-tomoko-d@city.yokote.lg.jp; 6Omori Municipal Hospital, Yokote 013-0525, Japan; onot@oomorihp.jp

**Keywords:** social frail, older adults, depression, COVID-19 pandemic

## Abstract

Background: Recent longitudinal studies have reported proportion of frailty transition in older individuals during the COVID-19 pandemic. Our study aimed at clarifying the impact of social frailty in community-dwelling older adults during the COVID-19 pandemic and at identifying factors that can predict transition to social frailty. Methods: We performed this study from 2019 (before declaration of the state of emergency over the rising number of COVID-19 cases) to 2020 (after declaration of the emergency). We applied Makizako’s social frail index to our study subjects at the baseline and classified into robust, social prefrailty, and social frailty groups. Multiple logistic regression analysis was performed using robust, social prefrailty, or social frailty status as dependent variable. Results: Analysis by the Kruskal–Wallis test revealed significant differences in the score on the GDS-15 among the robust, social prefrailty, and social frailty groups (*p* < 0.05). Furthermore, multiple regression analysis identified a significant association between the social frailty status and the score on GDS-15 (odds ratio, 1.57; 95% confidence interval (95% CI), 1.15–2.13; *p* = 0.001). Conclusion: The increase in the rate of transition of elderly individuals to the social frailty group could have been related to the implementation of the stay-at-home order as part of the countermeasures for COVID-19. Furthermore, the increased prevalence of depressive symptoms associated with the stay-at-home order could also have influenced the increase in the prevalence of social frailty during the COVID-19 pandemic.

## 1. Introduction

The World Health Organization (WHO) reported cases of COVID-19 infection in China on 31 December 2019, declared a Public Health Emergency of International Concern (PHEIC) on 30 January 2020, and declared the COVID-19 pandemic on 11 March 2020. The Japanese government declared a state of emergency on 16 April 2020. Additionally, it announced three major preventive measures against COVID-19: avoidance of closed spaces, crowded places, and close-contact settings (also called the “Three Cs”). Other important preventive measures included physical distancing, use of face masks, handwashing, stay-at-home notice, and restrictions on social gatherings [1,2]. While living through COVID-19 countermeasures, many people experienced drastic changes in their ordinary lifestyles, including fearfulness of contracting COVID-19 infection, depression and excessive sleep [3,4,5,6]. The drastic changes associated with the implementation of COVID-19 countermeasures have been implicated in the high incidence of “Corona-Frailty” and recent longitudinal studies have reported a high proportion of frailty transition in older individuals during the COVID-19 pandemic [7,8,9]. Shinohara et al. (2021) reported a prevalence of physical frailty (or physical prefrailty) of 8.8% (or 52.1%) in community-dwelling older adults living under COVID-19 countermeasures [10]. Although there have been follow-up studies on the transition to physical frailty, longitudinal investigations of social frailty, including a decline in social activities or social participation have not been well documented. With a rapidly aging society in Japan social frailty is a serious issue for community-dwelling older adults in this era of the COVID-19 pandemic. As a preventive measure during the COVID-19 pandemic, community organizations have closed. Older adults are constrained from visits with friends, therefore the social participation have been restricted [11]. Recently studies reported that the prevalence of anxiety and depression among the general population during the pandemic. Furthermore, social isolation, loneliness and depression have, in turn, been associated with cognitive decline and incident dementia among older adults [12]. In recent years, social frailty has been defined provisionally as the absence of social resources, social activities, and self-management abilities for fulfilling basic social needs [13,14]. Additionally, Makizako et al. (2015) reported assessing the social frailty index based on responses to simple questions relating to going out less frequently as compared with the previous year, visiting friends sometimes, feeling helpful towards friends or family, living alone, and talking with someone every day [15]. We do not therefore know to what extent stringent social distancing during the COVID-19 pandemic might have affected the changes in the rate of conversion to social frailty. Thus, we hypothesized that the prevalence of social frailty in rural community-dwellers would have greatly influence on limitation of social activities due to the COVID-19 pandemic. The current study was part of a nationwide clinical registry in Japan called the Organized Registration for the Assessment of dementia on Nationwide General consortium toward Effective treatment (ORANGE) [16]. The clinical stage focused on the ORANGE consists of preclinical, MCI, early-stage dementia and advanced-stage dementia. Throughout a longitudinal investigation for individuals in Japan, their lifestyle, social background, genetic risk factors, and required care level are assessed. Thus, we believe that this cross-sectional study would lead to better understanding of preclinical stages of dementia and improvement of effective care management towards dementia for Japan’s ageing population. Therefore, our study was aimed at clarifying the impact of social frailty in community-dwelling elder persons during the COVID-19 pandemic and factors predictive of a social frailty status.

## 2. Materials and Methods

### 2.1. Participants

Participants were recruited from six areas of Yokote city in Akita prefecture, Japan. Our study was performed from 2019 (before declaration of the state of emergency over the rising number of cases of COVID-19) to 2020 (after declaration of the state of emergency). The participants were aged 65 years old or over, had the ability walk independently, and were living at home without personal assistance. The exclusion criteria were dementia, major depression, severe hearing or visual impairment, stroke, Parkinson’s diseases, other neurological disease, intellectual disability, need for support or care as certified by the Japanese public long-term care insurance system due to disability, and inability to complete cognitive tests at the baseline assessment. A total of 161 participants were recruited at the baseline survey (before declaration of the state of emergency), classified into 103 persons in a robust state, 37 persons in a social prefrail state, and 11 persons in a social frail state. For the analyses in this study, the 103 participants in a robust health state were followed up after one year (after declaration on the state of emergency).

### 2.2. Criteria for the Diagnosis of Social Frailty

The previous studies aimed to gain a better understanding of social frailty among community-dwelling older adults by exploring the relationship between selected determinants of social frailty and functional decline. In examining the risk of future need for support and need for care, using measures that assess social aspects of frailty (e.g., solitary life and relationship with others), five aspects of social frailty were found to be associated with new incidence of need for support and long-team care within 24 months [13]. Makizako’s social frailty index was used for the assessment in this study and consisted of the following questions: (i) living alone (yes), (ii) talking with someone everyday (no), (iii) feeling helpful to friends or family (no), (iv) going out less frequently compared with last year (yes), and (v) visiting friends sometimes (no). Subjects with a 0 were classified as being in the robust group, those with a score of 1 as being in the social prefrailty group, and those with a score of 2–5 as being in the social frailty group. According to the change in the social frailty index at the end of the follow-up period, persons classified as being in the R group at the baseline were grouped into three groups at the end of the follow-up period [13].

### 2.3. Assessment and Outcome

We evaluated the physical performance by measuring the grip strength (kg) and usual walking speed (m/s). Each participant was also assessed by four cognitive subtests of the National Center for Geriatrics and Gerontology Functional Assessment Tool (NCGG-FAT) [17], and touch panel-type dementia assessment scale (TDAS) [18] was also applied in this study. The detailed information on each cognitive subtests indicated as the follow.

#### 2.3.1. Components of NCGG-FAT

The NCGG-FAT is the computerized multidimensional neurocognitive test on an iPad (Apple, Cupertino, CA, USA) with a 9.7-inch touch display for cognitive screening in population-based samples. The outcome of the NCGG-FAT enables assessment of the effects of intervention on multidimensional cognitive function among older adults [17]. The task instructions were presented with a letter size of at least 0.0001 m^2^ on the display. The overall criterion is determined from the standard deviation of the measured values based on each average value. The judging criteria are as follows, 1; lowest, 2; slightly lower, 3; normally, 4; fine, and 5; very fine. For this study, a trained operator supported each participant by setting up the tablet PC and running each test. Participants completed the NCGG-FAT subtests as follows (Figure 1).

Task 1: Tablet Version of Word Recognition

This test comprised the two computerized tasks of immediate recognition and delayed recall. In the first task of immediate recognition, participants were instructed to memorize 10 words, each of which was displayed for 2 s on the tablet PC. After that, a total of 30 words, including 10 target and 20 distracter words were shown to the participants, and they were required to select the 10 target words immediately. Three trials of this task were repeated. The average number of correct answers were assigned a score in the range of 0 to 10. In the second task, the participants were asked to correctly recall the 10 target words after 20 min. The number of correctly recalled target words was assigned a score in the range of 0 to 10. Finally, we calculated the sum of the scores on the two tasks of immediate recognition and delayed recall. 

Task 2: Trail Making Test Version A and Task 3: Trail Making Test Version B

In the Trail Making Test Version A (TMT-A) task, participants were instructed to touch the target numbers in a sequence as rapidly as possible. Target numbers from 1 to 15 were randomly displayed on the tablet panel. The Trail Making Test Version B (TMT-B) required the participants to touch target numbers (e.g., 1–15) and letters in turn. The required time (seconds) to complete each task was recorded, within a maximum time of 90 s.

Task 4: Symbol Digit Substitution Task

As the fourth test, in the Symbol Digit Substitution Task (SDST), nine pairs of numbers and symbols were shown in the upper part of the tablet display. A target symbol was shown in the center of the tablet panel, and selectable numbers were displayed at the bottom. Participants were asked to touch the number corresponding to the target symbol shown in the central part of the tablet display as rapidly as possible. The number of correct numbers within 120 s was recorded.

#### 2.3.2. Components of TDAS

The TDAS test is a sensitive and comprehensive assessment battery for rating the symptoms of Alzheimer’s disease. The hardware for the TDAS comprises a 14-inch touch panel display and computer devices. The TDAS operates using Windows OS, and was developed with reference to the Alzheimer’s disease assessment scale-cognitive subscale (ADAS-cognition). The TDAS subtests consisted of seven of the ADAS-cognition test items and two other tasks, and can typically be administered within 30 min. For the participants, each subtest of the TDAS was instructed verbally or visually by the computer. A score of the TDAS ranged from 0 for a perfect score to 101 for all incorrect answers. The judging criteria are as follows, ≦6; normal, 7–13; dementia prevention area, and ≧14; suspected dementia. The TDAS subtests indicated as follows (Figure 2).

Task 1: WR

The Word recognition test was a computerized test based on the Word recognition task of ADAS-cognition. At the start of this task, 12 target words were individually presented on the display for 3 s each at 2 s intervals. After demonstrating the target words, the computer randomly displayed 24 words consisting of 12 target words and 12 non-target words. Participants were then instructed to respond by touching the displayed button of “yes,” “no,” or “unknown” in response to the question regarding whether the word had been shown previously. Participants completed the trial three times. The total number of incorrect responses for three trials was recorded (maximum score = 72).

Task 2: Following a Command

This task was modified from the command task of ADAS-cognition. The computer presented 10 selectable icons labelled from 0 to 9 and then required participants to touch the number specified. An incorrect response is recorded as one point by the computer. The number of incorrect responses in two trials was scored (maximum score = 2).

Task 3: Orientation

This task was based on the orientation task of ADAS-cognition. The computer displayed four screens in sequence. On each screen, participants were asked to touch selectable icons and answer what year, month, day, and weekday it is. The number of incorrect responses was scored (maximum score = 4).

Task 4: Visual-spatial perception task

This task was modified from the constructional praxis task of ADAS-cognition to evaluate visual-spatial perception. The ADAS-cognition requires subjects to copy the geometric forms presented. As it is difficult for a computer to assess the accuracy of a drawn form automatically, this test examines whether the subject can remember geometric forms correctly. The computer first presented four screens displaying a target geometric form (i.e., a square, rhombus, cube, or triangular prism) for 5 s each. Participants were then required to correctly select the target form in response to a question task including the target form and four non-target forms. The number of incorrect responses was scored (maximum score = 4).

Task 5: Naming fingers

This test assessed whether the participants could name the fingers correctly, using the protocol of ADAS-cognition. Participants were asked to correctly respond to a picture question of a hand marked with a red circle, by touching an icon labelled with the five finger names. An incorrect response was assigned a score of 1 (maximum score = 5).

Task 6: Object recognition

This task was based on the naming objects task of ADAS-cognition for computerization, examines whether the subject can recognize the utilization of tools correctly. For this task, the computer presents three screens in turn. Participants were instructed to touch the correct usage icon (e.g., a pair of scissors, comb or broom) among five selectable icons labelled with the purpose of usage. Three trials were performed, and an incorrect response was assigned a score of 1 (maximum score = 3).

Task 7: Accuracy of order of a process

This task was modified from the ideational praxis of ADAS-cognition, examines whether the subject can recognize the process of writing a letter through to posting it. The computer displayed seven icons labelled randomly with seven actions. Participants were asked to correctly touch the icons in order. The number of incorrect responses was recorded (maximum score = 5).

Task 8: Money Calculation

This task assessed the money calculation ability of each participant. Participants needed to combine coins equal to an amount of money from various denominations of coins displayed on the screen. Three trials were performed, and an incorrect response was scored as one point (maximum score = 3).

Task 9: Clock Time Recognition

This task included three kinds of questions regarding clock time recognition. The computer asks three kinds of questions. First, the subject is asked to select one clock from a choice of six that shows the same time presented on the screen. Second, the subject is asked to select one icon from a choice of six that labels the time corresponding to a clock presented on the screen. Third, the subject is asked to select one icon from a choice of six that labels the difference in time between a clock presented on the screen (shows 10:40) and 11 o’clock. An incorrect response is scored as one point (maximum = 3).

### 2.4. Statistical Analysis

Age, medication status, education levels, and the scores on the UWA, GS, WM, TMT-A, TMT-B, SDST, TDAS, and GDS-15 were analyzed using the Kruskal–Wallis test to compare the baseline data among the R, SPF, and SF groups. Chi-squared test for 2 × 2 contingency was applied for gender.

Next, the proportion of persons classified into each of the three groups based on the social frailty index at the end of the follow-up period was calculated. To determine the factors associated with the social frailty status, we applied multiple logistic regression analysis using the robust, social prefrailty, or social frailty statuses as a dependent variable. As independent variables, age, medication, education, and the scores on the UWS, GS, WM, TMT-A, TMT-B, SDST, TDAS, and GDS-15 at the baseline were entered into the regression model. The robust group was set as the reference group for analysis (robust group = 1; social prefrailty group = 2; social frailty group = 3). SPSS Version 27.0 for Windows (SPSS INC., Chicago, IL, USA) was used for the analysis, and the level of a significance was set at *p* = 0.05.

## 3. Results

According to Makizako’s social frailty index, the 103 participants in the robust group at the baseline (before declaration of the state of emergency) were classified at the end of the study period into the robust group (*n* = 63), social prefrailty group (*n* = 29), or social frailty group (*n* = 11) (Figure 3). Table 1 lists the demographic data of the participants. The Kruskal–Wallis test was used to analyze the differences in the characteristics of the participants among the three groups, and revealed significant differences in the GDS-15 score among the groups (*p* < 0.05), and more depressive symptoms in the social frailty group. However, there were no differences in the subject characteristics, such as the age, gender, medication status, education level, or scores on the UWS, GS, WM, TMT-A, TMT-B, SDST, or TDAS at the baseline among the groups.

Next, the proportion of patients in each social frailty domain at the end of the follow-up period is shown in Table 2. The proportions (%) of subjects providing answers to each of the questions included in the social frailty index assessment were as follows; social frailty group (*n* = 11): 0% for “Living alone (% yes)”, 9.1% for “Talking with someone everyday (% no)”, 100% for “Feeding helpful to friends or family (% no)”, 27.3% for “Going out less frequently compared with last year (% yes)”, and 100% “Visiting friends sometimes (% no)”, social prefrailty group (*n* = 29): 6.9% for “Living alone (% yes)”, 6.9% for “Talking with someone everyday (% no)”, 0% for “Feeding helpful to friends or family (% no)”, 86.2% for “Going out less frequently compared with last year (% yes)”, and 0% “Visiting friends sometimes (% no)”, robust group (*n* = 63): 0% for each of the questions.

Finally, multiple regression analysis was performed to identify factors influencing changes in the social frailty status, which identified the score on GDS-15 as being significantly associated with transition to social frailty (odds ratio, 1.57; 95% confidence interval (95% CI), 1.15–2.13; *p* = 0.001) (Table 3).

IQR, interquartile range; UWS, Usual Walking Speed; GS, Grip Strength; WM; Word list Memory, TMT-A, Trail Making Test A version; TMT-B, Trail Making Test B version; SDST, Symbol Digit Substitution Task; TDAS, Touch Panel-type Dementia Assessment Scale; GDS-15, Geriatric Depression Scale.

UWS, Usual Walking Speed; GS, Grip Strength; WM; Word list Memory, TMT-A, Trail Making Test A version; TMT-B, Trail Making Test B version; SDST, Symbol Digit Substitution Task; TDAS, Touch Panel-type Dementia Assessment Scale; GDS-15, Geriatric Depression Scale.

## 4. Discussion

In this study, the subjects were classified based on Makizako’s social frailty index at the baseline (before the implementation of COVID-19 countermeasures) as follows: robust group, 64%; social prefrailty group, 23%; and social frailty group, 13%. The prevalence of social frailty in this study was higher than the rate reported by Tsutsumimoto et al. (11.1%) [19]. In this study, the participants were classified into a group that remained robust (*n* = 63, 61.1%), a group that converted from robust to social prefrailty (*n* = 29, 28.2%), and a group that converted from robust to social frailty (*n* = 11, 10.7%) during the one-year follow-up period. Although previous studies indicate that social activities, such as having conversations and going outdoors, increase stimulation of the brain and physical function [20], early during the COVID-19 pandemic, the Japanese government requested that the population avoid mass gatherings and practice strict social distancing. Makizako et al. (2013) previously reported that the prevalence of social frailty in community-dwelling older adults in Japan was 8.4% [21], and the transition rate from robust to social frailty (10.7%) in this study appeared to be higher than that reported by Makizako, which was based on a study conducted prior to the onset of the COVID-19 pandemic [15].

Next, we examined the physical performance and cognitive and mental function at the baseline among the three groups classified according to the social frailty status. We found that the score on GDS-15 was higher in the social frailty group than in the robust or social prefrailty group. Our result is in line with those of several previous studies, which revealed that higher levels of social frailty were associated with depressive symptoms [22,23]. In this study, we compared the GDS-15 score measured at the baseline and at the end of the one-year follow-up period, before the onset social frailty, and suggest, based on our results, that depressive symptoms may precede conversion to social frailty. On the other hand, contrary to expectation, there were no significant differences in the physical performance or cognitive functions among the groups. Previous studies have reported that the social aspects of the lives of older adults, including social engagement and the social environment, may impact the extent to which physical functions decline and physical inactivity increases [24,25]. Moreover, Makizako et al. reported that among independent community-dwelling older adults who are not physically frail, those who are socially frail may be at greater risk of developing physical frailty in the near future, and social frailty may precede (and lead to the development of) physical frailty. These could explain why there were no differences in the physical performance or cognitive functions among the groups in this study: however, the physical performance and cognitive functions could show accelerated decline in the near future.

Finally, multiple logistic regression analysis revealed that conversion to social frailty was significantly associated with depressive symptoms. This results is in line with those of several previous studies [22,26]. Recently, a longitudinal cohort study and a cross-sectional study reported associations between social frailty and depressive symptoms [22]. Several studies have reported behavioral changes, such as going out less frequently, reduced social contacts, and staying away from public places with the implementation of the COVID-19 emergency [27,28,29]. In particular, a deteriorated social life and fewer in-person social interactions observed during the pandemic were occasionally associated with an increased incidence of depression [30,31,32]. Theo et al. reported that among the Dutch community-dwelling older adults in May 2020, the average social loneliness increased slightly, and average emotional loneliness increased much more strongly during the COVID-19 pandemic [32]. In this study, declaration of the COVID-19 emergency was associated with decreased going out (27.2%), decreased desire of being helpful to friends or family (10.7%), and decreased visits to friends (10.7%). The Japanese government requested that the population avoid mass gatherings and practice strict social distancing during the COVID-19 pandemic, so that social interactions for community-dwelling older adults were canceled nationwide. In particular, older adults might have been more conscious of refraining because the infection is more likely to be severe. While social distancing is critical to prevent spread of the infection, it must be balanced with measures to prevent social frailty. Moreover, contacts between other people increases the spread of infection: however, even if the infection has been prolonged, maintaining meaningful social contact is of paramount importance. Recently, an online program for social communication in older adults was reported to be effective [33]. The National Center for Geriatrics and Gerontology is also making progress in providing opportunities for social participation and exercise through an online system for older adults [34]. In order to prevent increase in the incidence of social frailty, community-dwelling older adults should be supported while using these new online systems so as to maintain social interactions during the pandemic.

We need to consider the limitations of our research before we design future research. First, our cohort was composed of a localized group of individuals from only one rural area of northern Japan, due to the difficulty in sampling and recruiting from a depopulated, small rural area. The impact of COVID-19 should be assessed according to the living environments of the subjects in order to identify more at-risk (e.g., infection status, population density). Second, we could not arrive at any conclusions about the cause-and-effect relationship between stay-at-home orders during the emergency and the occurrence of depressive symptoms. The effect of the stay-at-home orders on the incidence of future depressive symptoms still needs to be clarified.

## 5. Conclusions

In our study, the participants transitioned from robust to social frailty (10.7%) during the one year after state of emergency. An increase in social frail might indicate effect by stay home of the implementation of COVID-19 countermeasures. Furthermore, we suggest that depressive symptoms are affected by staying home, increasing the prevalence of social frailty.

## Figures and Tables

**Figure 1 ijerph-19-00986-f001:**
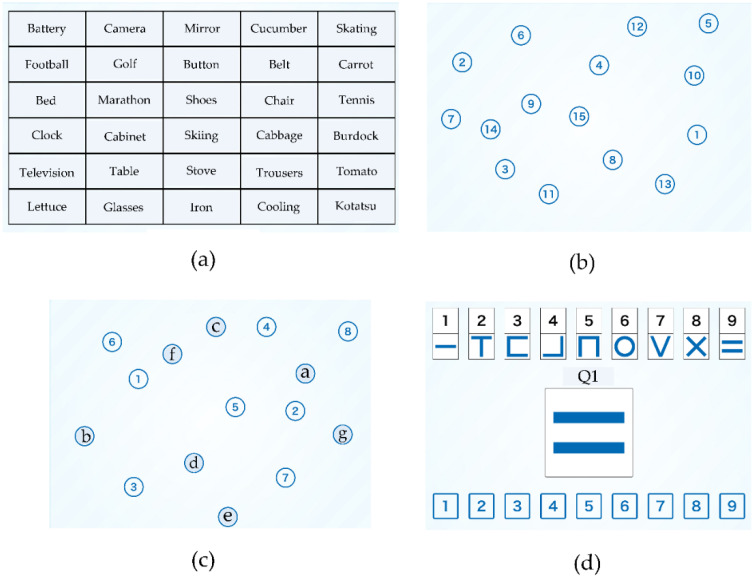
Sample of NCGG-FAT. (**a**) Task 1: Tablet Version of Word list Memory. (**b**) Task 2, Tablet Version of Trail Making Test Version A. (**c**) Task 3, Tablet Version of Trail Making Test Version B. (**d**) Task 4: Symbol Digit Substitution Task.

**Figure 2 ijerph-19-00986-f002:**
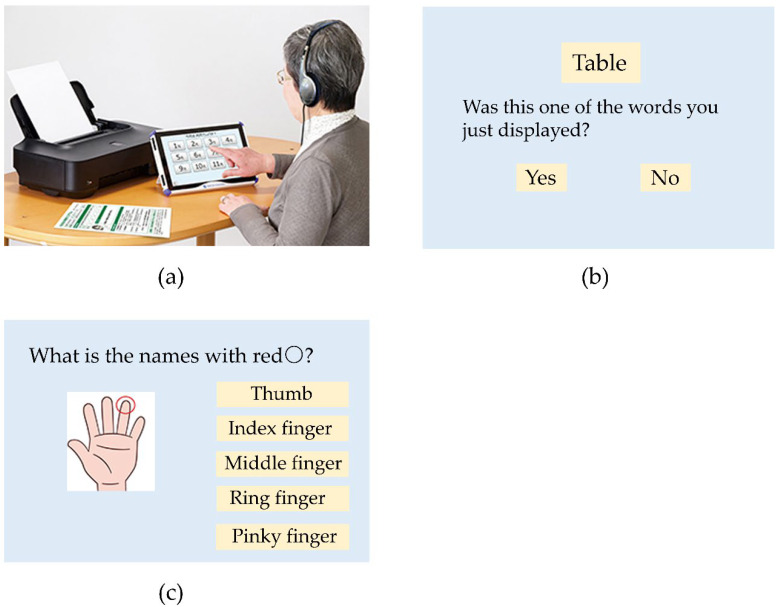
Sample of ADAS. (**a**) The hardware for the TDAS. (**b**) Task 1, Word recognition. (**c**) Task 5, Naming Fingers.

**Figure 3 ijerph-19-00986-f003:**
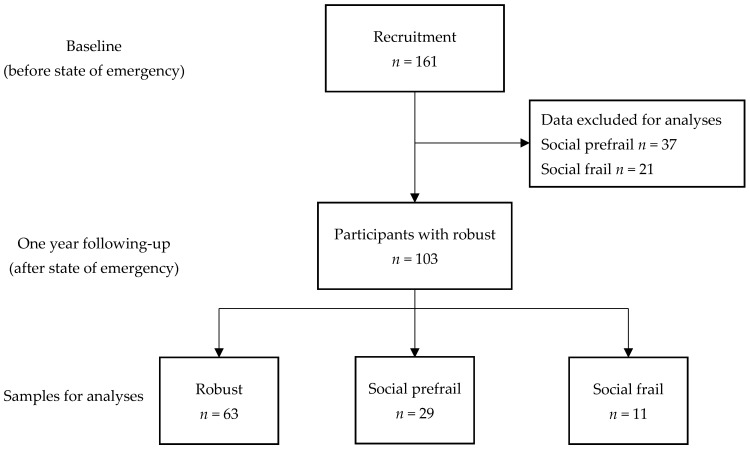
Flow-chart of participants’ selection for analyses.

**Table 1 ijerph-19-00986-t001:** Characteristics of participants with robust (*n* = 103) at baseline.

Status	Robust		Social Prefrailty		Social Frailty		*p* Value
	*n* = 63		*n* = 29		*n* = 11		
	Median	IQR	Median	IQR	Median	IQR	
Age (years)	73.5	8.0	74.5	10.0	71.0	8.0	0.380
Gender (*n*)	Male 46, Female 17		Male 20, Female 9		Male 6, Female 5		0.464
Medication (*n*)	2.0	4.0	3.0	4.0	3.5	2.0	0.265
Education (years)	12.0	3.0	12.0	3.0	12.5	3.0	0.363
UWS (m/s)	1.16	0.27	1.15	0.31	1.11	0.19	0.528
GS (kg)	23.6	8.7	24.7	8.8	24.9	10.3	0.898
WM (score)	12.7	4.3	11.7	5.3	13.0	6.3	0.170
TMT-A (s)	20.0	7.0	24.0	12.0	19.5	6.0	0.306
TMT-B (s)	37.5	27.0	40.0	21.0	33.0	15.0	0.454
SDST (score)	41.5	14.0	38.0	24.0	41.5	11.0	0.341
TDAS (score)	3.0	4.0	3.0	6.0	4.5	6.0	0.266
GDS-15 (score)	2.0	2.0	2.0	3.0	6.0	6.0	0.021 *

* *p* < 0.05, Kruskal–Wallis test.

**Table 2 ijerph-19-00986-t002:** Percentage of each social frail domain at the follow-up period.

	Robust	Social Prefrailty	Social Frailty	Total
Number of the participants (*n*)	63	29	11	103
Living alone (% yes)	0	6.9	0	1.9
Talking with someone everyday (% no)	0	6.9	9.1	2.9
Feeling helpful to friends or family (% no)	0	0	100	10.7
Going out less frequently compared with last year (% yes)	0	86.2	27.3	27.2
Visiting friends sometimes (% no)	0	0	100	10.7

A value indicates percentage of each social frail state at the following-up period.

**Table 3 ijerph-19-00986-t003:** Result of Multiple logistic regression analysis.

Groups	Social Prefrail (*n* = 29)				Social Frailty (*n* = 11)			
	Odds	95%CI		*p* Value	Odds	95%CI		*p* Value
Age (years)	0.96	0.86	1.08	0.53	0.94	0.77	1.14	0.54
Gender (female/male)	3.30	0.60	1.82	0.17	3.25	0.22	1.47	0.39
Medication (*n*)	1.01	0.85	1.20	0.90	1.11	0.82	1.51	0.50
Education (years)	0.97	0.75	1.27	0.84	1.27	0.80	2.04	0.31
UWS (m/s)	0.37	0.02	5.68	0.48	0.89	0.06	6.53	0.31
GS (kg)	0.93	0.84	1.05	0.24	0.95	0.80	1.14	0.59
WM (score)	0.83	0.67	1.04	0.11	1.23	0.86	1.74	0.26
TMT-A (s)	1.05	0.96	1.14	0.31	0.96	0.78	1.19	0.73
TMT-B (s)	0.98	0.94	1.02	0.23	0.98	0.92	1.04	0.45
SDST (score)	0.99	0.92	1.06	0.73	0.98	0.88	1.09	0.69
TDAS (score)	0.91	0.78	1.07	0.25	1.20	0.95	1.51	0.12
GDS-15 (score)	0.91	0.71	1.18	0.48	1.57	1.15	2.13	0.001 *

Reference group for analysis was robust group (e.g., robust group = 1; social prefrailty group = 2; social frailty group = 3 for each category of dependent variables). For a nominal scale of the gender, a dummy variable of Female = 0 or Male = 1 was defined for statistics. Likelihood-ratio test, * *p* < 0.05; Nagelkerke’s R^2^ = 0.346.

## Data Availability

No publicly archived datasets, analyzed or generated, were used in this study.

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
