# Peer review of "Impact of COVID-19 Pandemic Exacerbation of Depressive Symptoms for Social Frailty from the ORANGE Registry"

_ijerph, 2022, doi:10.3390/ijerph19020986_

Round 1

Reviewer 1 Report

Your contribution “Impact of COVID-19 pandemic exacerbation of depressive symptoms for social frailty from the ORANGE Registry” is an important and interesting one for International Journal of Environmental Research and Public Health.

The introduction leads generally to the investigation. Relevant literature is used. Theoretical backgrounds, e.g., theoretical derivations of the tasks and test, could be highlighted more.

In the methodological part, the participants of the study and the criteria for the selection of subjects were described in a comprehensible way. The criteria for social frailty were central to this. A variety of tasks were described to provide insight. Although the authors provided examples of tasks, the bullet points from 2.3 could be presented more clearly. Perhaps a sample task could also be illustrated on the iPad or display.

More detailed information could be provided for data analysis, e.g. Was the use of the tablet or the display logged by means of log data during work? The log data could be used, for example, to check the extent to which the subjects with different initial or background characteristics differ in their usage behavior. In addition, information about the time taken to complete the respective task could be added. Did the subjects receive direct feedback on their respective success or failure in completing the tasks? Etc.

After the methodological part, the authors went directly to the results. So far, it has not been possible to identify a research question in the article. This should definitely be made up for.

Please also check your presentation of results. e.g. "N" means, as far as I know, the size of the population and "n" the number of characteristic values. Is there something mixed up in your presentation?

 „According to a result of Kruskal-Wallis test, 195 GDS-15 has significant difference among the groups (p < 0.05).“ Can they elaborate on that, please?

 The results were discussed in detail and conclusively using appropriate literature.

The bibliography must be checked and errors corrected.

Author Response

Response to Reviewer 1 Comments

Point 1: The introduction leads generally to the investigation. Relevant literature is used. Theoretical backgrounds, e.g., theoretical derivations of the tasks and test, could be highlighted more.

Response 1: Each cognitive task used in the present study has been already standardized with some reports of validation or reliability for NCGG-FAT or TDAS. Thus, we have explained clearly about details of each task with some citations. Furthermore, we have added the detail for significance of this study. (in red)

(We do not therefore know to what extent the stringent social distancing during the COVID-19 pandemic might have affected the changes in the rate of conversion to social frailty. Thus, we hypothesized that the prevalence of social frailty in rural community-dwellers would have greatly influence on limitation of social activities due to the COVID-19 pandemic.Page2, Line64)

Point 2: In the methodological part, the participants of the study and the criteria for the selection of subjects were described in a comprehensible way. The criteria for social frailty were central to this. A variety of tasks were described to provide insight. Although the authors provided examples of tasks, the bullet points from 2.3 could be presented more clearly. Perhaps a sample task could also be illustrated on the iPad or display.

Response 2: Thank you for supportive comments. As the reviewer suggested, we have added more information on criteria of social frailty or each cognitive task, and we have added figure on the ipad display. (Page3)

(The previous studies aimed to gain a better understanding of social frailty among community-dwelling older adults by exploring the relationship between selected determinants of social frailty and functional decline, In examining the risk of future need for support and need for care, by using measures that assess social aspects of frailty (e.g., solitary life and relationship with others), 5 aspects of social frailty were found to be associated with new incidence of need for support and long-team care within 24 months [13]. Page3, Line95)

Point 3: More detailed information could be provided for data analysis, e.g. Was the use of the tablet or the display logged by means of log data during work? The log data could be used, for example, to check the extent to which the subjects with different initial or background characteristics differ in their usage behavior. In addition, information about the time taken to complete the respective task could be added. Did the subjects receive direct feedback on their respective success or failure in completing the tasks? Etc.

Response 3: Thank you for supportive comments. Outcomes of each cognitive test (e.g. TMT-A, TMT-B, SDST) over the NCGG-FAT or TDAS apply the log data during testing on the application of iPad. Also, to properly control subjects’ usage behavior in each cognitive task, an instruction and a practical task for each test are demonstrated on a tablet’s display before performing each test of NCGG-FAT or TDAS. In addition, the participants have completed the respective task, and we applied outcomes (e.g. correct responses or required times) to complete the each cognitive task. Regarding the last reviewer’s comment, each participant directly received feedback using a report paper after completed all the tasks of NCGG-FAT or TDAS. Therefore, the above contents have been added in the section of “Methods”.

Point 4: After the methodological part, the authors went directly to the results. So far, it has not been possible to identify a research question in the article. This should definitely be made up for.

Response 4: Thank you for supportive comments. We have added our hypothesis in the section of “Introduction”. Furthermore, we have added and rewritten about "Result".

(Thus, we hypothesized that the prevalence of social frailty in rural community-dwellers would have greatly influence on limitation of social activities due to the COVID-19 pandemic. Page2, Line66)

Point 5: Please also check your presentation of results. e.g. "N" means, as far as I know, the size of the population and "n" the number of characteristic values. Is there something mixed up in your presentation?

Response 5: Thank you for supportive reviews comments . As suggested by the reviewer, the presentation of “n” has been unified throughout the manuscript.

Point 6:  „According to a result of Kruskal-Wallis test, 195 GDS-15 has significant difference among the groups (p < 0.05).“ Can they elaborate on that, please?

Response 6: Thank you for supportive reviews comments. As we showed the description of “GDS-15 were analyzed using the Kruskal-Wallis test to compare the baseline data among robust, social prefrail and social frail group.” in the section of data analysis, a statistical result of GDS-15 has been elaborated according to a result of the Kruskal-Wallis test.

(The Kruskal-Wallis test was used to analyze the differences in the characteristics of the participants among the three groups, and revealed significant differences in the GDS-15 score among the groups (p < 0.05), and more depressive symptoms in the social frailty group. Page5, Line239)

Point 7: The results were discussed in detail and conclusively using appropriate literature.

The bibliography must be checked and errors corrected

Response 7: Thank you for supportive reviews comments. According to reviewers suggestion, we have that the bibliography checked and errors corrected.

Reviewer 2 Report

The authors proposed “Impact of COVID-19 pandemic exacerbation of depressive symptoms for social frailty from the ORANGE Registry”. However, there are certain flaws in this paper that must be addressed.

Some of the concerns are as below:

  • Firstly, systematic numbering for contents and sub-contents is Section 2. Methodology is confusing; it is preferable to number subsections using the alphabet, such as a, b, and so on, rather than a numeric number. For example, in section 2.3. Assessment and outcome, "Cognitive subtests indicated as the following," However, subsections the numbering starts with 2.4, which is perplexing. Thus, subsections must use the alphabets or 2.3.1, so on.
  • The ORANGE Registry must be explained for general understanding.
  • Section 2.3. Assessment and outcome, “Four cognitive subtests of the National Center for Geriatrics and Gerontology Functional Assessment Tool (NCGG-FAT)”, based on the current presentation, appears to have only three subtests. Whereas Components of TDAS look like new components with 11 test items. To my understanding, TDAS is one of four cognitive subtests, and its components are further subdivided into 11 test items. Please double-check and renumber the items.
  • The article should be formally written rather than using short acronyms, such as
  • 2, line 92, The task instructions were presented with a letter-size of at least 1.0 × 1.0 cm2 on the display, instead of cm2, it would be better to write as a square meter.
  • 3, line 98, “10 words, each of which was displayed for 2 s on the tablet PC, instead of “s” rewrite as seconds.
  • 3, line 103 “…….after 20 min”, “min” as minutes so on.
  • Summaries of the methodology components used in the article should be presented in a graphical format for easier reading.
  • What exactly is ADAS-cog? Is it a case of ADAS-cognition? If that's the case, please don't use acronyms.
  • In Table 1. Characteristics of participants with robust (N = 103) at baseline, why is it that female gender participant are included while male gender participants are excluded? Furthermore, in Table 3, gender mentioned for male is 1 and female is 0, this content is confusing. The number of genders considered in each group should be mentioned clearly.
  • Comparative study of social frail between the genders could add good impact to this article.

Author Response

Response to Reviewer 2 Comments

Point 1: Firstly, systematic numbering for contents and sub-contents is Section 2. Methodology is confusing; it is preferable to number subsections using the alphabet, such as a, b, and so on, rather than a numeric number. For example, in section 2.3. Assessment and outcome, "Cognitive subtests indicated as the following," However, subsections the numbering starts with 2.4, which is perplexing. Thus, subsections must use the alphabets or 2.3.1, so on.

Response 1: Thank you for a supportive comment. As suggested by the reviewer, we have revised naming of subsections.

Point 2: The ORANGE Registry must be explained for general understanding.

Response 2: Thank you for supportive reviews comments. As suggested by the reviewer, we have added description of the Orange Registry, with the below reference.

(The current study was part of a nationwide clinical registry in Japan called the Organized Registration for the Assessment of dementia on Nationwide General consortium toward Effective treatment (ORANGE) [14]. The clinical stage focused on the ORANGE consists of preclinical, MCI, early-stage dementia and advanced-stage dementia. Throughout a longitudinal investigation for individuals in Japan, their lifestyle, social background, genetic risk factors and required care level are assessed. Thus, we believe that this cross-sectional study would lead to better understanding of preclinical stages of dementia and improvement of effective care management towards dementia for Japan's ageing population. Page2, Line68)

Reference) Saji N., Sakurai T., Suzuki K., et al. ORANGE investigators ORANGE's challenge: developing wide-ranging dementia research in Japan. Lancet Neurol 2016, 15(7):661-662. doi: 10.1016/S1474-4422(16)30009-6.

Point 3: Section 2.3. Assessment and outcome, “Four cognitive subtests of the National Center for Geriatrics and Gerontology Functional Assessment Tool (NCGG-FAT)”, based on the current presentation, appears to have only three subtests. Whereas Components of TDAS look like new components with 11 test items. To my understanding, TDAS is one of four cognitive subtests, and its components are further subdivided into 11 test items. Please double-check and renumber the items.

Response 3: Thank you for a supportive comment. We have reconstituted the description of NCGG-FAT or TDAS (Page3, Line118, with a reference as follow;

Reference for TDAS) Masashi I., Daiki J., Miyako T., Katsuya U. Touch Panel-type Dementia Assessment Scale: a new computer-based rating scale for Alzheimer's disease: Psychogeriatrics 2011, 1(1), 28-33, doi: 10.1111/j.1479-8301.2010.00345.x.

Reference for NCGG-FAT) Makizako H., Shimada H., Park H., Doi T., Yoshida D., Uemura K. et al. Evaluation of multidimensional neurocognitive function using a tablet personal computer: test-retest reliability and validity in community-dwelling older adults, Geriatr Gerontol Int 2013, 13, 860-866, doi.org/10.1111/ggi.12014.

Point 4: The article should be formally written rather than using short acronyms, such as

line 92, The task instructions were presented with a letter-size of at least 1.0 × 1.0 cm2 on the display, instead of cm2, it would be better to write as a square meter.

3, line 98, “10 words, each of which was displayed for 2 s on the tablet PC, instead of “s” rewrite as seconds.

3, line 103 “…….after 20 min”, “min” as minutes so on.

Response 4: Thank you for supportive reviews comments. According to reviewer’s suggestion, we have revised and changed as following.

(The task instructions were presented with a letter size of at least 0.0001 m2 on the display. Page3, Line122)

(In the second task, the participants were asked to correctly recall the 10 target words after 20 minutes. Page4, Line137)

Point 5: Summaries of the methodology components used in the article should be presented in a graphical format for easier reading.

Response 5: Thank you for a supportive comment. As suggested by the reviewer, we have revised the methodology components, and we have added figure on the ipad display (Page3).

Point 6: What exactly is ADAS-cog? Is it a case of ADAS-cognition? If that's the case, please don't use acronyms.

Response 6: Thank you for supportive reviews comments. According to reviewer’s suggestion, we have revised and changed “ADAS-cog” to “ADAS-cognition” as follows.

(The TDAS test is a sensitive and comprehensive assessment battery for rating the symptoms of Alzheimer's disease. The hardware for the TDAS comprises a 14-inch touch panel display and computer devices. The TDAS operates using Windows OS, and was developed with reference to the Alzheimer's disease assessment scale-cognitive subscale (ADAS-cognition). The TDAS subtests consisted of seven of the ADAS-cognition test items and two other tasks, and can typically be administered within 30 minutes. Participants were instructed verbally or visually by the computer to complete the TDAS subtests as follows.

Page4, Line156)

Point 7: In Table 1. Characteristics of participants with robust (N = 103) at baseline, why is it that female gender participant are included while male gender participants are excluded? Furthermore, in Table 3, gender mentioned for male is 1 and female is 0, this content is confusing. The number of genders considered in each group should be mentioned clearly.

Response 7: Thank you for a supportive comment. As the participants (n = 103) includes female and male, we have changed the indication of gender in Table 1. Additionally, a dummy variable regarding gender has been definitely indicated at a footnote in Table 3.

Point 8: Comparative study of social frail between the genders could add good impact to this article.

Response 8: Thank you for a helpful comment. According to a result of the multiple logistic regression analysis, significant difference of the gender wasn’t highlighted. So that, comparative study of social frail between the genders would be included in the next research.

Round 2

Reviewer 1 Report

Thank you for your revision.

The reviewers' comments have been implemented, which has improved the quality of the article. I still have the following recommendations: 

  1. The theoretical background of the paper could be described in more detail. Which theoretical approaches do you refer to besides the current state of research?
  2. In the methodological part there are many enumerations, which could be presented even more clearly. This would be clearer especially with the different tasks.

Author Response

Response to Reviewer 1 Comments

Point 1: The theoretical background of the paper could be described in more detail. Which theoretical approaches do you refer to besides the current state of research?

Response 1: Thank you for supportive comments. We have added our hypothesis in the section of “Introduction”.

(As a preventive measure during the COVID-19 pandemic, community organizations have closed. Since older adults are constrained from visits with friends, the social participation have been restricted [11]. Recently some studies reported that the high prevalence of anxiety and depression among the general population during the pandemic [12,13]. Furthermore, social isolation, loneliness and depression have been associated with cognitive decline and incident dementia among older adults [14]. Page2, Line58)

Point 2: In the methodological part there are many enumerations, which could be presented even more clearly. This would be clearer especially with the different tasks.

Response 2: Thank you for supportive comments. As the reviewer suggested, we have added more information on criteria of social frailty or each cognitive task, and we have added figure  1 and Figure 2.

Reviewer 2 Report

Although the authors addressed some of the concerns raised earlier, there are still a few more issues that need to be addressed before publishing.

  • Figure 1 is hazy, and it would be best to keep the resolution at 300 dpi. For ease of understanding, the Japanese characters in the figure should be translated into English.
  • A period should be placed at the conclusion of the figure caption. The figure numbering is incorrect; instead of being written as Figure 2, it is drafted as Figure 1.
  • In the Table 3 caption, there should be a space between the Table and the number "3". Align Table 3 such that the word "Odds" may be viewed clearly.
  • Line 263, “A value indicates percentage ….”. Remove the space after the table. Follow the same for Line 266, “Reference group for analysis…”.
  • Double-check the references and make any required changes in accordance with the journal template. Furthermore, please follow the same format as others; some have a space between the author's initial and middle name. They don't have in refs. 5, 31, and so on.
  • Please double-check the word spacing and indent alignment.

Author Response

Response to Reviewer 2 Comments

Point 1: Figure 1 is hazy, and it would be best to keep the resolution at 300 dpi. For ease of understanding, the Japanese characters in the figure should be translated into English.

Response 1: Thank you for a supportive comment. As suggested by the reviewer, we have revised Figure 1.

Point 2:  A period should be placed at the conclusion of the figure caption. The figure numbering is incorrect; instead of being written as Figure 2, it is drafted as Figure 1.

Response 2: Thank you for supportive reviews comments. As suggested by the reviewer, we have modified "Figure 1" to "Figure 2". (Page 7, Line 278)

Point 3: In the Table 3 caption, there should be a space between the Table and the number "3".

Response 3: Thank you for a supportive comment. We have added a space between the Table and the number "3".  (Page 8, Line 287)

Point 4: Align Table 3 such that the word "Odds" may be viewed clearly.

Response 4: Thank you for supportive reviews comments. According to reviewers suggestion, we have revised Table 3.

Point 5: Line 263, “A value indicates percentage ….”. Remove the space after the table. Follow the same for Line 266, “Reference group for analysis…”.

Response 5: Thank you for a supportive comment. As suggested by the reviewer, we have remove the space after the table.

Point 6: Double-check the references and make any required changes in accordance with the journal template. Furthermore, please follow the same format as others; some have a space between the author's initial and middle name. They don't have in refs. 5, 31, and so on.

Response 6: Thank you for supportive reviews comments. As suggested by the reviewer, we have revised references.
